# Using the Reach Effectiveness Adoption Implementation Maintenance (RE-AIM) Framework to Evaluate a Tailored Education Program to Reduce Obesity-Related Cancers in El Paso, Texas

**DOI:** 10.3390/ijerph21081051

**Published:** 2024-08-09

**Authors:** Jennifer J. Salinas, Roy Valenzuela

**Affiliations:** Department of Social Work, College of Health Sciences, University of Texas at El Paso, El Paso, TX 79912, USA; rvalenzuela12@utep.edu

**Keywords:** latinos/hispanics, obesity, cancer, prevention, education, intervention, Pasos Para Prevenir Cancer, steps to prevent

## Abstract

**Background**: Pasos Para Prevenir Cancer is a tailored lifestyle program that focuses on behavioral modification through knowledge and behavioral strategy education, which was delivered in El Paso, Texas, on the U.S.–Mexico border. **Methods:** Using the RE-AIM framework, we assessed Pasos Para Prevenir Cancer for efficacy and potential for sustainability. Survey, administrative, and observational data were collected between 2018 and 2022. The program was evaluated to determine reach, effectiveness, adoption, implementation, and maintenance. **Results:** Tailoring and adapting to the U.S.–Mexico border context is feasible and necessary to deliver evidence-based healthy eating and active living education content. Pasos Para Prevenir Cancer was well received and delivered in diverse settings with varying linguistic needs. Components of the program were adopted by other organizations and integrated into existing programming. **Conclusions:** Adapting and tailoring evidence-based programs to improve healthy eating and active living is required to meet the needs of Latino subgroup populations, like those on the U.S.–Mexico border.

## 1. Introduction

Cancer is the leading cause of death in the Mexican American population residing in the Texas–Mexico border region [1,2]. While the incidence of screening-eligible cancers is generally lower than other non-Hispanic or non-border populations, the border population carries a disproportionate burden of cancer deaths that could be prevented with screening or lifestyle interventions [3]. Obesity is associated with 13 different types of cancers [4] and is a leading modifiable risk factor for cancer among Mexican Americans [5]. Nearly 40% of Mexican Americans living on the Texas–Mexico border are obese or overweight [6].

Implementation of evidence-based obesity prevention programs could impact the current cancer burden in this highly susceptible population. Lifestyle education is a principal component of any obesity prevention or mitigation program [7]. Participation in evidence-based programs leads to sustained changes in healthy eating and active living (HEAL) behaviors and weight loss [8,9,10,11,12,13,14,15,16,17,18,19,20,21,22,23,24,25,26,27,28,29,30,31,32,33,34,35]. There have been two main strategies for education interventions implemented in Mexican American populations. First, numerous programs are designed to address behaviors that affect Mexican American children and their families [9,11,14]. Strategies are family-based to mitigate childhood obesity and prevent adult onset of chronic diseases. The second approach targets adults at key life course stages to reverse early signs of or manage chronic diseases in middle to older age. Programs like the DPP, DASH, ¡Vivir Mi Vida!, Su Corazón Su Vida, The Latino Health Project, Mujeres Fuertes Y Corazones Saludables, BAILA, Proyecto Mamá, and the culturally adapted ‘Health Dads, Healthy Kids’ are burgeoning evidence-based programs [16,20,24,33]. These programs have been delivered across the United States to improve HEAL behaviors among Mexican Americans and other Latino adults at various stages of the life course that have demonstrated successful mitigation or management of chronic diseases.

Despite the effectiveness of interventions that use strategies to improve HEAL behaviors, there exists a gap in knowledge about the effectiveness of programs that are intended to change behaviors that increase the risk for cancer among Latinos. Moreover, available approaches have not been widely tested in the U.S.–Mexico border region. Nearly 3 million people live on the Texas–Mexico border [36]. Most of these residents are of Mexican origin and account for approximately one-third of all Latinos living in the state [36]. Socioeconomic conditions on the border, such as food access or walkability, serve as barriers to a HEAL lifestyle and serve as social determinants of health, driving health disparities, including cancer types observed in this region [37,38,39].

There are a plethora of evidence-based options available to address behaviors related to obesity in the U.S.–Mexico border region [16,20,24,34]. However, few have been designed to address the unique context of the border. Therefore, tailoring and adaptation are essential, even in programs that have been tested in other Latino populations. Therefore, dissemination of effective strategies to this region requires consideration of the unique needs of the target population. Pasos Para Prevenir Cancer is an obesity-related cancer prevention program that was adopted using evidence-based strategies and tested in El Paso, Texas, on the U.S.–Mexico border region. The program is designed to provide evidence-based curricula tailored to meet the educational needs of this unique population [40]. To date, more than 5000 people have been educated on obesity-related cancers, obesity, goal setting, nutrition, and physical activity. Participation in the program leads to sustained behavioral change [41]. 

In this paper, we present findings from our program evaluation of the Pasos Para Prevenir Cancer program. We use the Reach Effectiveness Adoption Implementation Maintenance (RE-AIM) framework to evaluate adaptation, tailoring, and implementation strategies and assess potential dissemination and scalability. While most culturally tailored programs use linguistics, tailoring to ethnic food or other cultural needs of the target community, Pasos Para Prevenir Cancer was implemented to reach the diverse socioeconomic and cultural preferences of the U.S.–Mexico border region. Therefore, we describe these unique challenges and how we addressed them. 

## 2. Materials and Methods

### 2.1. Study Design

We used administrative data, surveys, and observational data for this evaluation study. Administrative data came from administrative records used to track attendance and session location. Additionally, we collected survey data from participants who participated in at least one session and agreed to complete surveys at baseline, 6 months, and 12 months. Participants were contacted by phone or email and invited to participate, and if agreed, were sent an email with a link to the informed consent. If consent were obtained, participants would receive a link to complete the baseline survey. Surveys would be sent out automatically at 6 months and 12 months. Follow-up phone calls were made for no-response surveys, and participants were offered the chance to complete the survey with staff over the phone. We collected data on 300 participants. Observational data were collected from quarterly reports provided to the funding agency. This generally included signs of adoption or maintenance of program components by community partners and agencies.

### 2.2. Intervention

Pasos Para Prevenir Cancer was a tailored intervention to prevent obesity-related cancer [40]. We reviewed multiple bilingual obesity programs for content and strategies. Core components across programs were adopted and tailored to the U.S.–Mexico border context. Key elements included Body Mass Index (BMI), goal setting, healthy eating on a budget, and finding a physical activity right for you. Community advisors provided feedback on curriculum components, and further tailoring to the curriculum was made based on recommendations. The final PPPC program curriculum included a total of 5 sessions offered based on the interest and needs of individual community organizations. All material was available in English and Spanish, and sessions were delivered by trained educators or community health workers who were bilingual. 

### 2.3. Measurements of the Five RE-AIM Dimensions

Table 1 provides an overview of the RE-AIM components and how they were measured in our evaluation. Each component’s data and evaluation process are described below. 

#### 2.3.1. Reach

Program reach was evaluated using attendance logs to determine program participation and trends over time. This included both education and resource navigation. We did not collect data on refusals due to our large-scale outreach efforts that included mass emails and outreach to organizations. 

#### 2.3.2. Effectiveness

Participants for this evaluation study were recruited from the PPPC program. We collected data on physical activity using the International Physical Activity Questionnaire (IPAQ) [39]. Nutrition was evaluated using the Food Frequency Questionnaire (FFQ) [40]. Weight and Body Mass Index (BMI) data were collected using self-report due to contact restrictions associated with COVID-19. 

#### 2.3.3. Adoption

We tracked session offerings, location, and attendance numbers. We used these data to determine the number and types of organizations that agreed to host our programming. Additionally, we noted changes or policy efforts made by each organization in our quarterly reports to our funder. Changes that occurred varied from regularly hosting our education program and purchasing food for participants to engage in cooking demonstrations to starting regular exercise groups for participants and employees. 

#### 2.3.4. Implementation

A number of changes were required to implement the curriculum material in the El Paso context. Iterations were documented and tracked quarterly. We then assessed what iteration works best under what circumstance and determined any trends or most likely circumstances when one strategy would be used over another [37].

#### 2.3.5. Maintenance

We determined maintenance at the organizational and individual levels. Organizational maintenance was assessed by efforts to incorporate our programming into existing offerings. For example, the curriculum can be added to scheduled educational programs that occur regularly. Additionally, creating innovative programs based on our tailored strategies (i.e., cooking or exercise classes). Maintenance at the individual level was determined by sustained behavioral change related to diet or physical activity [38].

### 2.4. Data Analysis

Data were entered into Excel or REDCap depending on type (administrative, survey, observational). Data were then cleaned and assessed for irregularities. Specific analyses are described below. 

#### 2.4.1. Administrative Data

Administrative data were typically entered into Excel after each education session was conducted. Data collected included date, location, and number of participants. Location was categorized into type (e.g., school, church, employer, city, etc.). Comparisons were made to determine what type of locations had the most participants and the most sessions. Additionally, analysis was conducted to assess temporal patterns over time. 

#### 2.4.2. Survey Data

Survey data were collected using REDCap (Nashville, TN, USA). Data were cleaned, and then frequency and distribution analyses were conducted using STATA 16 SE (College Station, TX, USA). Analysis was conducted to compare physical activity and nutrition patterns over time to determine change between baseline and 6 months and signs of sustainability between 6 months and 12 months. A detailed description of the methodology can be found elsewhere [37].

#### 2.4.3. Observational Data

Observational data were collected quarterly as part of a report process to the funder. This information was assessed for trends and common themes such as new locations or partners, partners who added our program to an existing program, and instances where the partners used our programming to start their own educational program. When common themes or conceptualizations became apparent, they were coded under the same classification. 

## 3. Results

### 3.1. Reach

Table 2 presents the number of participants who took part in our education program by modality. PPPC educated 8619 participants in total. Social media ‘live’ sessions were the largest percentage of participants due to COVID-19 and the wide reach that social media had during that time (*n* = 4710, 55%). Despite the pandemic, we were able to deliver a high number of in-person classes (2976, 34%) and less than expected participation through webinars (993, 11%). Although not seen in the table, it is important to mention that we navigated 5223 people who also participated in our education programming. Additionally, we did not collect demographic data on participants since we used a simple sign-in sheet to obtain attendance numbers for each session. Participants that were recruited from the sessions to our evaluation study were, however, 87% female, primarily between the ages of 18 and 59, married and high school graduates. This is a conservative number, given our wide reach on social media and ability to reach a larger volume of participants with resource information. 

### 3.2. Effectiveness

Using our survey data, we analyzed participant physical activity, nutrition, and weight trends between baseline and 6 months to assess change and between 6 months and 12 months to evaluate sustained change. We observed a meaningful change between baseline and 6 months in physical activity engagement that was sustained at 12 months. These results were previously published and reported [38]. We found that participants who were immigrants, Spanish-speakers, and those with less than a high school education were the most responsive in improving physical activity engagement. We also observed promising results that are under review at present. Approximately a third of participants made the effort to increase their intake of healthier foods. Changes included a shift from higher fat and sugar content consumption to light and low-fat foods and greater consumption of ground chicken, lean red meat, and seafood.

### 3.3. Adoption

Table 3 provides an overview of the types of organizations that we delivered and how many sessions. Some of the most active sites included schools, employers, senior centers, the housing authority, and recreation centers. Locations that were less likely to offer their locations for classes included churches and clinics. This may be due to COVID-19 restrictions in general. Since the pandemic, we have observed an uptick in requests from our Oncology clinic, suggesting that there may have been more clinic interest, but due to COVID-19 restrictions, we did not receive as many referrals as we would likely have received.

### 3.4. Implementation

In most cases, we held multiple sessions over the course of the program, most often at schools (36% with two or more series sessions), employers (50% with two or more series sessions), and the Housing Authority (29% with two or more series sessions) (see Table 3). Part of the success at these locations was the co-offering approach that we took. In these locations, there were existing mandates or programming schedules already in place. Parents, residents, and employees were already familiar with schedules or the type of programming offered during scheduled sessions or received benefits from going. We observed this pattern in other settings as well. For example, at senior centers, our sessions were often scheduled around ‘bingo’ or other group activities that drew older adults to the centers. 

### 3.5. Maintenance

Table 4 presents successful strategies that led to the maintenance of components of PPPC. There were four implementation strategies used that were the most successful. First, we had a diverse network of community partners who provided locations for education program delivery. Partners ranged from senior centers and parks to schools and the housing authority. Tailoring allowed us to adapt to the needs of varying locations and demographics. Second, when possible, we provided collaborative programming that complemented existing activities already provided by our partners. For example, one partner, Live Active El Paso, offered programming in the parks or recreation centers where we would provide education or a cooking demo (See Table 4). We would also collaborate with them to promote their activities as part of our navigation offering. Social media and web-based platforms like Zoom were critical in obtaining information about session times and dates, as well as providing live education during COVID-19, navigating resources in the area, or demonstrating a cooking recipe that was highlighted during our sessions. Finally, navigation took on various forms beyond flyers and handouts. We hosted live events on social media with demonstrations, hosted group events with local city, county, and state parks, and worked with private businesses to host demonstrations such as yoga or nutrition at local parks and other public spaces (see Table 4). These activities led to regular use of resources by community members and planned events offered by community partners.

## 4. Discussion

Pasos Para Prevenir Cancer is a tailored and adapted program that uses evidence-based content and strategies delivered to the high-risk community in El Paso, Texas, located in the U.S.–Mexico border region. We used evidence-based education programming tested on Hispanic populations outside of the U.S.–Mexico border region and adapted it to the specific needs and interests of our target population. We tailored our program to fit the individual needs of our partners and complement existing programs. Participants demonstrated sustained change in physical activity engagement and nutrition. We delivered our program to 215 different sites ranging from schools to senior centers. Since few programs have been tested on the U.S.–Mexico border region, tailoring evidenced-based programs to this context is feasible and effective, and doing so may help address the persistent health disparities that exist in this region.

### 4.1. Using Community Partners to Tailor to Population Needs

Developing community partnerships to help adapt evidence-based programs to the specific needs of target communities is essential to effective implementation [41,42]. Key to our success in reaching such a diverse number of community organizations was working with our partners to understand the community’s needs and tailor program content to reach different demographic groups. Community organization representatives provided us with feedback on curriculum content, number of sessions, demonstration content, resources, and recipes for cooking demonstrations. From this feedback, we developed a menu of options for organizations to choose from and were able to accommodate the needs of varying settings. This menu of options allowed us to deliver across contexts such as employers, school districts, and senior centers. 

### 4.2. How to Involve Participants

Involving participants and keeping them engaged in any education program depends largely on trial and error. We would use feedback from participants, organization representatives, and attendance logs. We used this information to create our menu and add or remove content that was needed or of no interest to our participants. For example, we learned many programs focused on Choose My Plate. Because of this overlap, we modified our program to focus on solutions to barriers to eating healthy. Additionally, we added demonstrations and information about ‘super foods’ to tailor to learners already with basic nutritional knowledge. We used what was working, where there seemed to be interest and used activities that improve participant engagement and likelihood of sustaining behavioral change. 

### 4.3. Opportunity for Sustainability

Working with community partners to align your program with their needs or existing programs is the most effective approach for PPPC implementation in El Paso, Texas. These same approaches provide vehicles for sustainability through strengthening partnerships and developing agency capacity to deliver healthy eating and active living programming. We leveraged existing community organization structures and programming to deliver PPPC. By doing so, we complimented and enhanced the organization’s community engagement efforts. For example, many school districts provide parent education on a monthly or weekly basis. Our educational sessions became a regular component of this programming. We are now training the liaisons on how to implement the education program. 

### 4.4. Limitations

This evaluation was intended to critically assess the implementation of the Pasos Para Prevenir Cancer program in the unique context of the U.S.–Mexico border region. While the RE-AIM structure did help identify the strengths of the program implementation, we uncovered a few shortcomings that would need to be addressed as we move toward dissemination. First, we did not collect demographic data from the program participants. As we move forward with implementation, we should include these details on the sign-in sheets so we have information on the type of organization and the client population who attends our program. Second, there was not an evidence-based program on obesity-related cancer to draw from to inform that aspect of the intervention. While we filled a gap in culturally tailored interventions addressing cancer in a population that disproportionately is affected, more research is needed to understand how to best implement obesity-related cancer prevention programming. Finally, this program was implemented in an urban area, and future efforts should be made to implement it in rural areas to maximize dissemination potential.

## 5. Conclusions

Implementation of existing evidence-based programs in new settings requires flexibility and adaptability. While the curricula used in our program were tested on Hispanic populations, the U.S.–Mexico border context is unique in many ways. The population is primarily Mexican American but diverse in immigration experience, socioeconomic level, and educational attainment. This created new challenges in how to effectively implement PPPC. The COVID-19 pandemic created challenges and opportunities to use new strategies to deliver. Most importantly, working with community partners and being aware of their needs helps us create programming that would generate interest and meet the specific needs of our community. 

## Figures and Tables

**Table 1 ijerph-21-01051-t001:** RE-AIM Program Evaluation for Pasos Para Prevenir Cancer (PPPC).

RE-AIM Framework	Process Measures	Data Source(s)
**REACH:** Participation rate within the target population and characteristics of Participants compared to nonparticipants	# Participants in promotora and navigator programs.	Attendance sheets/sign-in records.
**EFFECTIVENESS:** Impact of the intervention on outcomes	Assessing the impact of educational programs on obesity knowledge and physical activity and nutrition behavioral change. Assessing the impact of the navigation program on utilization of community-based physical activity and nutrition resources. Assessing the capacity of the community partnership to maintain a system for resource sharing.	Pre-post surveys Community partner feedback Resource information contributions Updates and sharing of new resources
**ADOPTION:** Percent of community partners that are willing to adopt the intervention	# Community partners who agree to assist in educational programs. # Community partners that provide resource information for navigation guide. # Of community partners sharing information for web-based resource guide.	Community partner records/participation in our program Sign-up sheets Registration records Website maintenance records
**IMPLEMENTATION:** Can the program be consistently implemented?	# Physical activity and nutrition resources attended in the community. # Visits to web-based resource center. # Community partners participate in coalition.	Attendance sheets Registration records Social media insights
**MAINTENANCE:** The extent to which a program becomes institutional or a part of routine organizational practices and policies	# Community partners who adopt the programming. Sustained HEAL behavioral change.	Quarter reports Individual level sustained behavioral change.

**Table 2 ijerph-21-01051-t002:** Attendance by method of delivery plus navigation.

Attendance (*n* = 8619)	Total	% of Total Attendees
Online	933	11%
In-Person	2976	34%
Social Media	4710	55%

**Table 3 ijerph-21-01051-t003:** Program delivery locations and number of sessions at each site.

Organization	Total	1 Session	2 Sessions	≥3 Sessions
**Housing**	28 (13%)	20 (71%)	8 (29%)	0 (0%)
**Schools**	97 (45%)	62 (64%)	29 (30%)	6 (6%)
**Community Centers**	20 (9%)	10 (50%)	6 (30%)	4 (20%)
**Senior Centers**	18 (8%)	11 (61%)	5 (28%)	2 (11%)
**Employers**	34 (16%)	22 (50%)	17 (39%)	5 (11%)
**Churches**	6 (3%)	6 (100%)	0	0
**Clinics**	8 (4%)	2 (25%)	2 (25%)	4 (50%)
**Health Fairs**	4 (2%)	2 (50%)	2 (50%)	0 (0%)
**Location Totals**	215	135	69	21

**Table 4 ijerph-21-01051-t004:** Successful implementation strategies leading to maintenance of Pasos Para Prevenir Cancer.

Strategies	Description
**Diversity in community partner network**	PPPC expanded from primarily to local schools, senior centers, and employers to the YMCA’s, YWCA’s, El Paso Community College, El Paso Housing Authority, Churches, and city and county parks and recreation centers.
**Collaborative Programming**	Organizations such as Live Active El Paso, Communities in Schools, and El Paso Parks & Recreation Centers collaborated to provide shared activities that included PPPC programming. For example, El Paso Parks and Recreation Center’s ‘The Beast Recreation Center’, offered regular programming that included our cooking demonstrations to a live and online audience. We also worked with Live Active El Paso to promote shared programming that promoted walking and the use of city walking trails and paths.
**Social media and web-based platforms for delivery**	During the COVID-19 pandemic, we quickly moved our offering from in-person to online. We utilized Microsoft Teams, Webex, and Zoom to provide education to old and new community partners. We leveraged social media to reach new audiences while still meeting the needs of current partners. We also incorporated the use of a YouTube profile link that allowed anyone to access our educational content online.
**Navigation: From Paper to Practice**	Navigation in the form of recipes, grocery shopping lists, physical activity recommendations, and physical activity sites around the community was a staple to help maintain the knowledge learned during lessons. PPPC improved the use of navigation by hosting live physical activity events at locations provided in the navigation sheets, such as walking/running along the Playa Drain Trail, hiking The Franklin Mountains State Park, and hosting yoga and strength and conditioning sessions at local parks.

## Data Availability

The data used in this study will be made available upon email request to the corresponding author: Jennifer J. Salinas, PhD LMSW jsalinas7@utep.edu.

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
