# Peer review of "Using the Reach Effectiveness Adoption Implementation Maintenance (RE-AIM) Framework to Evaluate a Tailored Education Program to Reduce Obesity-Related Cancers in El Paso, Texas"

_ijerph, 2024, doi:10.3390/ijerph21081051_

Round 1

Reviewer 1 Report

Comments and Suggestions for Authors

This is an interesting manuscript that using the RE-AIM framework to evaulate the efficacy and sustainability of a tailored-education cancer preveiton program. The authors found that the tailored-education cancer prevention program can reach a diverse socio-economic and cultural preference populations. However, there are some points should be addressed and discussed in this manuscript. 

1. Please provide the baseline information of those subjects that participated in this program -- how about the age, gender, and SES effects on this program.

2. For Tables 2 and 3 -- Please provide age- and gender-specifgicaiton data for more information to evaualte the effectiveness of the program. 

3. The componentrs of RE-AIM could be demonstrate more detail for different study sub-population. 

4. The results of implementation and maintenance could also present with table formation. 

Author Response

Reviewer 1

This is an interesting manuscript that using the RE-AIM framework to evaulate the efficacy and sustainability of a tailored-education cancer preveiton program. The authors found that the tailored-education cancer prevention program can reach a diverse socio-economic and cultural preference populations. However, there are some points should be addressed and discussed in this manuscript. 

  1. Please provide the baseline information of those subjects that participated in this program -- how about the age, gender, and SES effects on this program.

We did not collect demographic data from our program because we used a sign in sheet to track attendance. However, we did collect data from the participants who volunteered for the evaluation study and added information about the demographics of our sample to page 5 lines 171 to 175.

  1. For Tables 2 and 3 -- Please provide age- and gender-specifgicaiton data for more information to evaualte the effectiveness of the program. 

Please see response for comment #1

  1. The componentrs of RE-AIM could be demonstrate more detail for different study sub-population. 

We agree with this comment and have added a limitation section that acknowledges this and other shortcomings to the study. See page 9, lines 281 to 295.

‘4.4 Limitations.

This evaluation was intended to critically assess the implementation of the Pasos Para Prevenir Cancer program in the unique context of the U.S.-Mexico border region. While the RE-AIM structure did help identify strengths of the program implementation, we uncovered a few shortcomings that would need to be addressed as we move towards dissemination. First, we did not collect demographic data from the program partici-pants. As we move forward with implementation, we should include these details on the sign-in sheets, so we have information on the type of organization, and the client population who attends our program. Second, there was not an evidence-based program on obesity related cancer to draw from to inform that aspect of the intervention. While we filled a gap in culturally tailored interventions addressing cancer, in a population that disproportionately is affected, more research is needed to understand how to best implement obesity-related cancer prevention programming. Finally, this program was implemented in an urban area and future efforts should be made to implement in rural areas to maximize dissemination potential.’

  1. The results of implementation and maintenance could also present with table formation. 

Thank you for this comment. Tables 3 and 4 present the implementation and maintenance findings. We realized that we did not reference table 3 in the text, so we added a reference on page 6, line 205. Additionally, table 4 presents information on strategies that helped promote maintenance. We added reference (i.e. see Table 4) throughout the maintenance findings description.

Reviewer 2 Report

Comments and Suggestions for Authors

Salinas + Valenzuela present their findings on Using the RE-AIM Framework to evaluate a tailored-education program to reduce obesity-related cancers in El Paso, Texas. Overall, it is a nice little paper which transparently presents how they implemented the program and some mitigation they did to adapt the delivery, to be participant centered.  I have some minor comments:

1. Keywords should include - PPPC and the English translation of Pasos Para Prevenir 'Steps to prevent'. 

2. Page 1 line 26 associated with 13 different types? of cancer

3. Line 41 page 1 should it be Healthy Dads, Healthy Kids? You have Health Dads 

4. Page 2 Line 54, including caners, observed in this region? Is that cancer rates? Types ect be clear. 

5. Page 3 line 96 it reads back on curriculum components and further tailored was made based on ....... This is not clear what are you saying the curriculum was tailored? Be clear.

Comments on the Quality of English Language

Overall, it is fine once the above points are made clear. I think it will read fine. 

Author Response

Reviewer 2

Salinas + Valenzuela present their findings on Using the RE-AIM Framework to evaluate a tailored-education program to reduce obesity-related cancers in El Paso, Texas. Overall, it is a nice little paper which transparently presents how they implemented the program and some mitigation they did to adapt the delivery, to be participant centered.  I have some minor comments:

  1. Keywords should include - PPPC and the English translation of Pasos Para Prevenir 'Steps to prevent'. 

Changes made.

  1. Page 1 line 26 associated with 13 different types? of cancer

Edits made

  1. Line 41 page 1 should it be Healthy Dads, Healthy Kids? You have Health Dads 

We are not clear on this comment. The text currently reads ‘adapted ‘Health Dads, Healthy Kids’ are burgeoning….’

  1. Page 2 Line 54, including caners, observed in this region? Is that cancer rates? Types ect be clear. 

Revised to ‘including cancer types observed in this region.’

  1. Page 3 line 96 it reads back on curriculum components and further tailored was made based on ....... This is not clear what are you saying the curriculum was tailored? Be clear.

We revised to read ‘Community advisors provided feedback on curriculum components and further tailoring to the curriculum was made’